# Effectiveness of web-based intervention for life-change adaptation in family caregivers of community-dwelling individuals with acquired brain injury: A cluster-randomized controlled trial

Yuka Iwata (Shindo)[1]*, Etsuko Tadaka[2]*

1 Department of Community Health Nursing, Graduate School of Medicine, Yokohama City University, Yokohama, Kanagawa, Japan, 2 Department of Community and Public Health Nursing, Graduate School of Health Sciences and Faculty of Medicine, Hokkaido University, Sapporo, Hokkaido, Japan

* t206705c@yokohama-cu.ac.jp (YI); e_tadaka@pop.med.hokudai.ac.jp (ET)

## Abstract

### Objective

To test the effectiveness of the web-based intervention "Koji-family.net 3-day program" (KF3 PGM) for life-change adaptation in family caregivers of community-dwelling individuals with acquired brain injury (ABI).

### Design

A cluster-randomized trial.

### Settings

All 82 institutions for families of individuals with ABI in Japan.

### Participants

Participants were 240 families at 16 different institutions for families of individuals with ABI. Inclusion criteria for participants were (1) families caring for an individual with ABI, (2) family members aged 20 years and over, and (3) the individual with ABI developed ABI when aged more than 16 years and less than 65 years.

### Methods

Clusters were randomly assigned to the intervention (8 clusters, $n = 120$) or the control (8 clusters, $n = 120$) group. For the intervention group, the KF3 PGM was assigned, in addition to routine family group activities to enhance the life-change adaptation. The control group followed their daily routine and received usual services. The primary outcome was the life-change adaptation scale (LCAS); secondary outcomes were the multidimensional scale of perceived social support (MSPSS) and the positive appraisal of care (PAC) scale at the

**Data Availability Statement:** All relevant data are within the article and its Supporting Information files.

**Funding:** This work was supported by the Sasakawa Scientific Research Grant from the Japan Science Society (grant number. 2021-6012). The funders had no role in study design, data collection and analysis, decision to publish, or preparation of the manuscript.

**Competing interests:** The authors have declared that no competing interests exist.

baseline, after 3 days (short-term follow-up), and after 1 month (long-term follow-up). A mixed model for repeated measures (MMRM) was applied.

## Results

A total of 91 participants were enrolled. The mean age (SE) of the participants was 64.0 (9.2) years; 87.8% of them were female. The intervention group showed better improvement in the LCAS than the control group in the whole study period ($F = 6.5$, $p = 0.002$). The mean observed change in LCAS from baseline was +8.0 (SE = 2.0) at 3 days and +11.6 (SE = 2.0) at 1 month in the intervention group ($F = 18.7$, $p < 0.001$). No significant differences in MSPSS and PAC were observed among the intervention and control groups in the whole study.

## Conclusions

The KF3 PGM can be an effective method of enhancing the adaptation to daily life in family caregivers of community-dwelling individuals with ABI. The results show that a potential web-based intervention in institutions for families of individuals with ABI plays a substantial, longer-term role in their support in Japan. Future studies could address the same research questions in different settings and cultures for family caregivers for even longer time periods.

## Introduction

Acquired brain injury (ABI) is an unforeseen condition with physical, cognitive, and psychosocial deficits that strongly affect a person's abilities [1–3]. In 2016, over 116.4 million individuals worldwide were estimated to suffer from ABI, and the prevalence has increased by 3.6% since 1990 [4–6].

Among individuals with ABI, 92.3% live with their families and receive care or assistance in their daily lives [7]. Because of the externally unrecognizable characteristics of ABI, individuals with ABI and their families are burdened with a lack of understanding and support in their community. Family caregivers of individuals with ABI are unique in causing semi-permanent, unexpected life changes such as role restructuring [8, 9], financial hardship [10], and loss [11, 12]. Our previous study focused on these unique life changes found that life-change adaptations are one of the benefits these individuals can achieve; we developed an instrument for measuring the life-change adaptation in family caregivers of individuals with ABI [13]. However, the development of programs to promote life-change adaptation has not yet been initiated globally.

Two points remain a challenge for intervention research: 1) The target subjects must be families of individuals with ABI in the community, and 2) the program should be indirect (web-based) rather than direct (face-to-face). In previous intervention studies over the past 10 years, the target subjects were recruited in a hospital [14–18] because most individuals with ABI receive long-term treatment at centralized rehabilitation centers that serve large geographic areas. Often overlooked is that the families of individuals with ABI play a substantial, longer-term role in their support after acute hospitalization [19]. Because of the community's lack of knowledge about ABI, no consensus has been reached as to how interventions should be provided after acute hospitalization. In addition, in previous intervention studies over the

past 10 years, the programs included educational interventions that could not be provided without direct, face-to-face participation by nurses, multidisciplinary professionals, or researchers [14–16, 18]. Often overlooked is the strain experienced by the families receiving the face-to-face program as a result of traveling long distances, which may lead them to forgo services altogether. Because of the lack of knowledge about web-based programs, no consensus has been reached as to how interventions should be provided indirectly. In Japan, the majority of community support for families is established through self-help groups for families of individuals with ABI. However, a unified program that includes web-based intervention has not been developed for these groups. Given the two aforementioned points, family institutions such as self-help groups for families of individuals with ABI can provide beneficial interventions delivered via online that eliminate barriers to treatment (e.g., time, distance), and the unavailability of knowledgeable providers for family caregivers in the community.

The model of stress and coping among caregivers is recommended as a useful model to enhance and explain adaptations to stressors in cases where the adaptations need to be practiced by family caregivers [20]. We firmly believe that this model is useful for planning interventions for enhancing life-change adaptations. This model is a theoretical model constructed from three domains (social support, appraisals, and coping) related to adaptation to stressors. Given the factors in life-change adaptation [13], interventions focused on the three aforementioned domains can trigger a change in the appraisal of caregiving resources / the health belief of life of family caregivers due to ABI for life-change adaptation.

The object of this research is to examine the effectiveness of a web-based intervention known as the Koji-fam.net 3-day program (KF3 PGM) for life-change adaptation of family caregivers of community-dwelling individuals with ABI. The hypothesis was that, compared with the usual service, KF3 PGM will improve life-change adaptation in family caregivers of community-dwelling individuals with ABI at 3 days (short-term follow-up) and 1 month (long-term follow-up). In this article, "individuals with ABI" refers to "disabled individuals with cognitive and behavioral dysfunction because of damage to the brain that occurs after birth and that is not related to a congenital disorder or a degenerative disease." "Family caregiver of individuals with ABI" refers to "a family relative that cares for or assists an individual with ABI in their daily lives."

## Materials and methods

### Study design, setting, and sample size

A cluster-randomized trial was used. Cluster randomization, instead of individual randomization, was adopted to reduce possible contamination among participants if an institution had both intervention and control group participants.

The unit of randomization is the institutions for families of individuals with ABI. All 82 institutions (cluster) for families of individuals with ABI in Japan were invited to participate in this study. Before clusters were randomly assigned to two groups, stratified randomization was adopted to reduce an unequal number of individuals that might be assigned to each arm of the study and was performed on the basis of a median cluster size 45 in an eligible cluster. Therefore, participants were then randomly assigned to two groups (ratio = 1:1)—8 clusters in the intervention group ($n$ = 120) and 8 clusters in the control group ($n$ = 120)—by independent research assistants using an SPSS syntax that generated a random number. The clusters mailed potential participants sequential invitations to express interest in joining the study. Both the intervention and control groups' participants were given a questionnaire for the survey via mail. All participants completed questionnaires at home. The survey was examined at three points in time: baseline, 3 days (short-term follow-up), and 1 month (long-term follow-up).

Baseline data were sent to the research institute (Department of Community Health Nursing, Yokohama City University; YCU) when the participant enrolled and completed the survey. In addition, short- and long-term follow-up data were sent to YCU by the end of the study. The collected data were entered by independent research assistants and managed by YCU. Data were collected from July 21 to October 30, 2021.

The sample size based on the following expectation was used. We powered the trial to have 80% power (alpha = 0.05; intraclass correlation coefficient (ICC) = 0.04) to detect a difference between groups on the primary outcome at follow-up to a standardized effect size (delta = 3.0; SD = 4.3). Allowing for 20% attrition, the estimated sample size was 4 participants per cluster; the required cluster size was 8 clusters per group. Our forecast was principally based on our pilot study findings (refer to S1 and S2 Appendices). The ICC was principally based on the findings of previous study estimates [21]. The number of participants per cluster was principally based on the findings of a previous study of family caregivers of individuals with ABI in Japan, where the average recruitment in the previous study was 4 participants per cluster [13].

## Participants

All 82 institutions for families of individuals with ABI in Japan were invited to participate in this study. Clusters' exclusion criteria were (1) less than 15 members, (2) unspecified members, and (3) refusals of explanations. Of the 18 eligible clusters, studies conducted in 16 randomly selected clusters were performed on the basis of the anticipated sample size. A total of 16 clusters and 240 members were enrolled in the study.

All 50 participants in each group expressed their participation in the study. Flow of the participants is described in Fig 1. Participants' inclusion criteria were (1) caring for an individual with ABI, (2) the caregiver was aged 20 years or more, (3) the individual with ABI developed ABI when aged more than 16 years and less than 65 years. The reasons for excluding cases in which ABI occurred in an individual aged more than 16 years or less than 65 years are that the adaptation process and outcomes for ABI are different [13]. Participants' exclusion criteria were (1) withdrew consent for data use and (2) missing LCAS baseline data.

## Intervention

The web- and community-based intervention (KF3 PGM) aimed to promote life-change adaptation. "Koji" in Japanese means "advanced dimension," covering everything outside the primary sensory areas and primary motor areas, including language, memory, attention, cognition, thought, and behavioral functions [22]. In this context, "koji" refers to individuals with brain dysfunction that interferes with advanced, complex, and abstract processing in ABI. KF3 PGM is formed by a theoretical framework based on the model of stress and coping among caregivers [20]. The KF3 PGM was developed in several iterative and small exploratory studies. First, the KF3 PGM prototype was developed through literature review. Second, the validity of the content of the KF3 PGM prototype was ensured through a focus group with 5 professionals who were selected from experienced researchers, clinical support staff, and family caregivers of individuals with ABI. Third, the KF3 PGM digital prototype was developed through interviews with 5 family caregivers of individuals with ABI who presented feedback on design and flow of the KF3 PGM digital prototypes.

The KF3 PGM, which is a fully automated web-based program, was offered to participants of intervention groups using the website. All the participants in the intervention group followed the same program: 5 to 15 minutes per content item, 3 consecutive days and 1 month of free daily practice. The 3 consecutive days of the KF3 PGM had three components: (1) a practical guide part, (2) an eco-mapping part, and (3) a message-board part. KF3 PGM is described

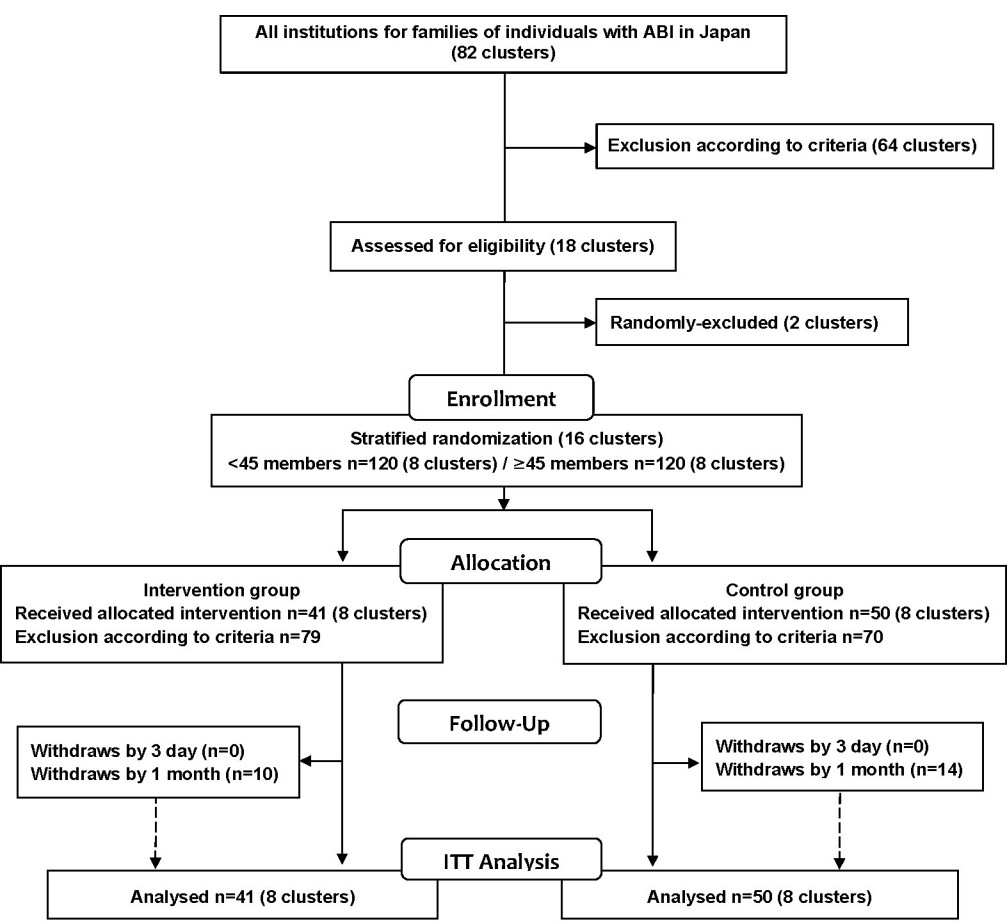

**Fig 1. Flow of the participants.**

ABI : acquired brain injury, LCAS: life change adaptation scale as primary outcome ITT: intention-to-treat

in Fig 2. After accessing the website, the participants were presented with three components. Participants watched a video explaining KF3 PGM and then worked on the other two components of KF3 PGM. Because KF3 PGM is a web-based intervention, there was no facilitator to guide the execution of KF3 PGM. After the 3 consecutive days of KF3 PGM, participants continued their eco-mapping and message-board participation efforts for 1 month. Participants received an invitation in the mail from an institutional staff member to access the dedicated KF3 PGM website.

## Comparison

The control group participants were guided in their daily routine. They were free to receive usual services such as consultation and information regarding ABI from institutions to which they belong. They were then placed on a waiting list to receive the intervention after 1 month.

## Outcomes and ethics

**Primary outcome.** The LCAS [13] was used to measure the life-change adaptation. "Life-change adaptation" refers to "the outcome of adaptation to changes in living resources / health belief of life of family caregivers due to ABI" [13]. It included 8 items. The LCAS scores range

**Intervention (KF3 PGM) group**

**Day 01**

**Practical guide** with website access
• Learn how to use the dedicated website
  through educational videos (5 min)

Preparation for activity

**Day 02**

**Eco-mapping** with website access
• Learn how to perform eco-mapping through video
  material (15 mins)
• Draw an eco-map of the six forms :
  1 Support for caregiving / 2 Mood / 3 Emergencies /
  4 Helping to understand ABI / 5 Respite / 6 Other

Reflection on social support

**Day 03**

**Message board** with website access
• Participate in the message board (15 min)
• Exchange information on four themes :
  1 Message of encouragement / 2 Acquired strengths /
  3 Self-promotion / 4 Needs for the community
• React to other participants' messages

Mutually beneficial networking

**Day 04 to 1 month**

**Daily practices**
• Log in to the official website
• Make an active commitment to the wider community

Reflection on social support

Mutually beneficial networking

**Study Period**

**Control group**

**Day 01 to 1 month**

**In daily routine**
**Receive usual services** regarding ABI from a family
institution to which they belong, as needed

**Fig 2. Intervention process in each group.**

from −40 to +40. The reliability of the scale has been established (Cronbach's α: 0.84), and validity was confirmed for Japanese participants in a previous study [13]. The Cronbach's α in the present study was 0.84 at baseline, 0.86 at 3 days, and 0.88 at 1 month.

**Secondary outcome.** The MSPSS Japanese version [23–25] was used to measure perceived social support. "Social support" refers to "an exchange of resources between at least two individuals perceived by the provider or recipient to be intended to enhance the well-being of the recipient" [23]. It included 12 items. The MSPSS scores range from 0.0 to 7.0. The reliability of the scale has been established (Cronbach's α: 0.91), and the validity was confirmed for Japanese participants in a previous study [25]. The Cronbach's α in the present study was 0.93 at baseline, 0.92 at 3 days, and 0.95 at 1 month.

The PAC [26] was used to measure positive appraisal of care. "Positive appraisal of care" refers to "any cognitive and affective evaluation of one's caregiving experience that describes caregiving as good and comfortable" [27]. It included 21 items. The PAC scores range from 0 to 100. The reliability of the scale has been established (Cronbach's α: 0.92), and the validity was confirmed for Japanese participants in a previous study [27]. The Cronbach's α in the present study was 0.91 at baseline, 0.91 at 3 days, and 0.90 at 1 month.

**Demographic characteristics.** The participants' demographic characteristics included age, sex, and relationship to an individual with ABI. The demographic characteristics of the individuals with ABI included age, sex, age at the time of ABI occurrence, cause of ABI, and period after ABI.

**Ethics.** The study was approved by the Research Ethics Committee of the Yokohama City University on January 21, 2021 (No. A210100013-(1); see S1 and S2 Appendices). The informed consent document explained the voluntary nature of participation, management of data, and intention to publish the results. All participants received the informed consent document and were considered to have agreed to participate in this study if they made a check mark on the informed consent document. The trial is registered with the University Hospital Medical Information Network (UMIN) Center (No. UMIN000042463), and the authors affirm that all ongoing and related trials for this intervention are registered. UMIN has been recognized by the International Committee of Medical Journal Editors (ICMJE) as an acceptable registry (see http://www.icmje.org/about-icmje/faqs/clinical-trials-registration/). Results are presented in accordance with the CONSORT statement for randomized controlled trials (see S3 Appendix).

## Statistical analysis

The primary statistical hypothesis was that the LCAS scores would be improved in the intervention group compared with the control group at 3 days (short-term follow-up) and 1 month (long-term follow-up). The secondary statistical hypothesis was that the MSPSS and PAC scores would improve, as previously detailed.

Descriptive statistics such as the mean, standard deviation, and frequency distribution were used to describe the demographic characteristics of the participants. A mixed model for repeated measures (MMRM), which takes into account the multiple measurements (3 time points: baseline, short-, and long-term follow-ups), was performed. MMRM was used to test a group × time interaction. Fixed effects included 1) group allocation (intervention or control), 2) time, and 3) group × time interaction; random effects included cluster. Analyses were performed on an intention-to-treat basis and included all baseline respondents. Missing data were not explicitly imputed because MMRM applies with a restricted maximum likelihood solution to repeated measures analyses under the missing-at-random assumption. A $p$-value less than 0.05 was set as the significant level. IBM SPSS ver. 28.0 was used to analyze the data.

Data entry and data analysis were not blinded because a different coding system was assigned to different group participants (e.g., 1 for the intervention group and 2 for the control group). However, the person who managed the data did not collect the data in the field. Each cluster provided service to a target within their catchment area, and participants selected in the control and intervention groups were far from each other, enabling us to avoid information contamination.

## Results

### Participants' demographic characteristics

A total of 91 participants (intervention group $n = 41$; control group $n = 50$) were included in the intention-to-treat analysis (Fig 1). A total of 67 participants completed the study (retention rate: 26.4%): 24 participants withdrew by the 1 month interval (intervention group $n = 10$; control group $n = 14$). The mean age (SD) of the participants was 64.0 (9.2) years. A total of 87.8% of the participants were female. The mean age (SD) of individuals with ABI was 50.3 (11.8) years. A total of 83.5% of the individuals with ABI were male. The mean age (SD) at which the individual with ABI developed ABI was 36.3 (15.0) years. The mean (SD) of the care period after ABI was 13.6 (7.7) years. Table 1 shows participants' demographic characteristics at baseline in each group.

**Table 1. Demographic characteristics and outcome measures at baseline.**

| | | IV (*n* = 41) | | Cont (*n* = 50) | |
| --- | --- | --- | --- | --- | --- |
| | | Mean ± SD | | Mean ± SD | |
| | | *n* | (%) | *n* | (%) |
| **Family caregiver's characteristics** | | | | | |
| Age (years) | | 63.0 ± 9.3 | | 64.8 ± 9.0 | |
| Gender | Females | 39 | (95.1) | 40 | (81.6) |
| | Males | 2 | (4.9) | 9 | (18.4) |
| Relationship to ABI | Parent | 21 | (51.2) | 27 | (54.0) |
| | Spouse | 20 | (48.8) | 23 | (46.0) |
| Care period (years) | | 14.3 ± 7.2 | | 13.1 ± 8.1 | |
| K6 | | 5.1 ± 5.2 | | 6.8 ± 5.5 | |
| **Individuals with ABI characteristics** | | | | | |
| Age (years) | | 50.7 ± 12.3 | | 50.0 ± 11.5 | |
| Age at the time of ABI (years) | | 35.9 ± 15.6 | | 36.6 ± 13.7 | |
| Gender | Males | 36 | (87.8) | 40 | (80.0) |
| | Females | 5 | (12.2) | 10 | (20.0) |
| **Outcome measures** | | | | | |
| LCAS | | −2.0 ± 12.6 | | 0.7 ± 13.7 | |
| MSPSS | | 4.5 ± 1.2 | | 4.4 ± 1.3 | |
| PAC | | 48.3 ± 17.8 | | 46.5 ± 16.4 | |

Missing data were excluded.

IV: intervention group, Cont: control group, SD: standard deviation

K6: Depression (The Kessler 6; score range 0–24)

LCAS: Life-change adaptation (the life-change adaptation scale; score range, −40 to +40)

MSPSS: Perceived social support (the multidimensional perceived social support scale; score range, 0.0–7.0)

PAC: Positive appraisal of care (the positive appraisal of care scale; score range, 0–100)

### Change in the outcome measures for the intervention and control group

MMRM for testing the effect in the whole study period indicated significant group × time interactions for LCAS ($F = 6.5$, $p = 0.002$). The mean observed change from baseline in LCAS was +8.0 (SE = 2.0) at 3 days and +11.6 (SE = 2.0) at 1 month in the intervention group ($F = 18.7$, $p < 0.001$). No significant differences in MSPSS and PAC were observed among the intervention and control groups in the whole study. The mean observed change from baseline in MSPSS was +0.1 (SE = 0.1) at 3 days and +0.2 (SE = 0.2) at 1 month in the intervention group ($F = 1.2$, $p = 0.316$). Mean observed change from baseline in PAC was +2.8 (SE = 2.3) at 3 days and +6.5 (SE = 2.5) at 1 month in the intervention group ($F = 3.3$, $p = 0.041$) (Table 2).

## Discussion

This study demonstrates the efficacy of a web-based intervention, KF3 PGM, in improving life-change adaptation, perceived social support, and positive appraisal of care in family caregivers of community-dwelling individuals with ABI. The originality of this study is the use of cluster-randomization and measurement of the life-change adaptation by LCAS. The findings show that a potential web-based intervention in institutions for families of individuals with ABI plays a substantial, longer-term role in their support. That is, the unified web-based intervention in an institutional program has the potential to strengthen said institutions, which are

**Table 2. Changes in the outcome measures for the intervention and control groups.**

| | n | Group × time interactions[†] | | Mean observed change from baseline (SE)[‡] | | Intervention effect[‡] | | ICC |
|---|---|---|---|---|---|---|---|---|
| | | F | p value | 3 day | 1 month | F | p value | |
| LCAS | | 6.5 | .002** | .. | .. | .. | .. | 0.551 |
| Intervention | 41 | .. | .. | 8.0 (2.0) | 11.6 (2.0) | 18.7 | < 0.001** | .. |
| Control | 50 | .. | .. | 2.1 (1.4) | 2.7 (1.6) | 1.7 | n.s. | .. |
| MSPSS | | 0.5 | n.s. | .. | .. | .. | .. | 0.885 |
| Intervention | 41 | .. | .. | 0.1 (0.1) | 0.2 (0.2) | 1.2 | n.s. | .. |
| Control | 50 | .. | .. | 0.0 (0.1) | 0.1 (0.1) | 0.3 | n.s. | .. |
| PAC | | 1.2 | n.s. | .. | .. | .. | .. | 0.861 |
| Intervention | 41 | .. | .. | 2.8 (2.3) | 6.5 (2.5) | 3.3 | 0.041* | .. |
| Control | 48 | .. | .. | 1.4 (1.3) | 1.4 (1.4) | 0.8 | n.s. | .. |

[†]Result of a mixed model for repeated measures (MMRM) adjusted for fixed effects included 1) group allocation (intervention or control), 2) time, and 3) group × time interaction; random effects included cluster.

[‡]Result of a MMRM adjusted for fixed effects including time and random effects including cluster.

*$p < 0.050$

**$p < 0.010$

***$p < 0.001$

SE: standard error, n.s.: not significant

ICC: intraclass correlation coefficient

LCAS: Life-change adaptation (the life change adaptation scale; score range, −40 to +40)

MSPSS: Perceived social support (the multidimensional perceived social support scale; score range, 0.0–7.0)

PAC: Positive appraisal of care (the positive appraisal of care scale; score range, 0–100)

essentially self-help groups, and enhance life-change adaptation as an ABI family. In addition, this study is the first step toward looking at a self-help group that can build on caregivers' strengths such as life-change adaptation and examining a practicable unified program for a self-help group.

The intervention group participants exhibited better life-change adaptation than the control group in the whole study period. The improvement in cognitive outcomes by the 3-day program for 1 month of efficacy is similar to that reported by Barton and King [28]. This study confirms that the life-change adaptation of family caregivers of individuals with an ABI is as trainable by remote intervention as that of family caregivers of individuals with stroke [29]. Even for family caregivers of individuals with ABI, there is potential to promote their life-change adaptation through a personally implementable program.

Grounded in the model [20], the key to enhancing the adaptation was three domains: social support, appraisals, and coping [30–32]. In the case of social support, the MSPSS scores were not significantly different between groups. A previous study suggested that just involvement with the program can increase self-efficacy of attribution in the program population [33]. What both the intervention and control groups provide a family institution-based routine program might not affect each group in MSPSS. In addition, perceived social support has been detected in previous studies with significant changes at the 6-month post-survey, indicating the possibility of detecting changes over a longer period of time than in the present study [16, 34]. However, eco-mapping in the KF3 PGM is a tool that encourages reflection on the perceived social support [35, 36]. Thus, we found that the KF3 PGM served as a domain of social support that could facilitate life-change adaptation. In the case of appraisals, the PAC score improved from baseline to short- and long-term follow-up. A previous study suggested that

situations of interpersonal interaction can enhance the positive appraisal of care among family caregivers [37]. The KF3 PGM provides a message board facilitating interpersonal interaction. Thus, we found that the KF3 PGM served as a domain of appraisal that could facilitate life-change adaptation. Through two components of the KF3 PGM (eco-mapping and message board), social support and positive appraisal of the participants were enhanced.

Finally, the participation rate was similar to that of a previous study polling the same population [13]. Regarding the demographics of the family caregivers of individuals with ABI, the caregivers were mostly women (87.8%). The average age at which individuals with ABI developed ABI was 36.3 years (SD = 15.0). According to the official evaluation by the Japanese government and a previous study, this profile is nearly identical to the profile of participants in a survey on the family caregivers of individuals with ABI [7, 13]. The average period for care was 13.6 years (SD = 7.7), which shows that the population in this study reflecting the families of individuals with ABI plays a substantial, longer-term role in their support. Thus, the sample was deemed representative of the population of family caregivers of individuals with ABI.

## Study limitations

The first limitation concerns the study setting and/or culture; convenience sampling of institutions in Japan, rather than probability sampling in other settings and/or cultures, was used. In generalizing the findings of this study, one must consider that the study was conducted through institutions of family caregivers in Japan, where family caregiving and family ties are traditionally and relatively more important than in other cultures. The second limitation concerns the research methodology. Because the family caregivers were not blinded to the KF3 PGM assignment, the Hawthorne effect cannot be ruled out. Accordingly, future studies could address the same research questions for family caregivers in different settings and cultures and for even longer time periods.

## Supporting information

**S1 Appendix. Study protocol (English version).**
(PDF)

**S2 Appendix. Study protocol (Japanese version).**
(PDF)

**S3 Appendix. CONSORT 2010 checklist.**
(PDF)

**S1 Dataset.**
(XLSX)

## Acknowledgments

The authors express appreciation to all members of the Department of Community Health Nursing, Graduate School of Medicine, Yokohama City University, and all the family caregivers of individuals with ABI, who graciously gave of their time and energy to participate in this study.

## Author Contributions

**Conceptualization:** Yuka Iwata (Shindo), Etsuko Tadaka.

**Data curation:** Yuka Iwata (Shindo).

**Formal analysis:** Yuka Iwata (Shindo).

**Funding acquisition:** Yuka Iwata (Shindo), Etsuko Tadaka.

**Investigation:** Yuka Iwata (Shindo).

**Methodology:** Yuka Iwata (Shindo).

**Project administration:** Etsuko Tadaka.

**Resources:** Etsuko Tadaka.

**Software:** Yuka Iwata (Shindo).

**Supervision:** Etsuko Tadaka.

**Validation:** Yuka Iwata (Shindo), Etsuko Tadaka.

**Visualization:** Yuka Iwata (Shindo).

**Writing – original draft:** Yuka Iwata (Shindo).

**Writing – review & editing:** Yuka Iwata (Shindo), Etsuko Tadaka.

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
