## [Decision Letter · Decision Letter 0]

26 May 2022

PONE-D-21-37509Effectiveness of web-based intervention for life-change adaptation in family caregivers of community-dwelling individuals with acquired brain injury: A cluster-randomized controlled trialPLOS ONE

Dear Dr. Iwata(Shindo),

Thank you for submitting your manuscript to PLOS ONE. After careful consideration, we feel that it has merit but does not fully meet PLOS ONE’s publication criteria as it currently stands. Therefore, we invite you to submit a revised version of the manuscript that addresses the points raised during the review process.

Both reviewers signalled the need for major work in clarifying the methods and the result interpretation sections. Please address all their criticisms.  Please submit your revised manuscript by Jul 03 2022 11:59PM. If you will need more time than this to complete your revisions, please reply to this message or contact the journal office at plosone@plos.org. Please include the following items when submitting your revised manuscript:A rebuttal letter that responds to each point raised by the academic editor and reviewer(s). You should upload this letter as a separate file labeled 'Response to Reviewers'.A marked-up copy of your manuscript that highlights changes made to the original version. You should upload this as a separate file labeled 'Revised Manuscript with Track Changes'.An unmarked version of your revised paper without tracked changes. You should upload this as a separate file labeled 'Manuscript'.

We look forward to receiving your revised manuscript.

Kind regards,

Andrea Martinuzzi

Academic Editor

PLOS ONE

Journal Requirements:

This work was supported by the Sasakawa Scientific Research Grant from the Japan Science Society. 

Reviewers' comments:

Reviewer's Responses to Questions

**Comments to the Author**

1. Is the manuscript technically sound, and do the data support the conclusions?

Reviewer #1: Partly

Reviewer #2: Partly

2. Has the statistical analysis been performed appropriately and rigorously? 

Reviewer #1: No

Reviewer #2: Yes

3. Have the authors made all data underlying the findings in their manuscript fully available?

Reviewer #1: No

Reviewer #2: Yes

4. Is the manuscript presented in an intelligible fashion and written in standard English?

Reviewer #1: No

Reviewer #2: Yes

5. Review Comments to the Author

Reviewer #1: This is an interesting CRCT assessing the Effectiveness of web-based intervention for life-change adaptation in family caregivers of community-dwelling individuals with acquired brain injury.

The methods and materials section could be benefit with some structure in a way that flows better for readers. i.e. Trial design and participants, outcomes (include primary and secondary), randomisation, sample size (so expand – see point 3

1. The unit of clusters should be defined explicitly. It’s a bit confusing especially with the terminology of “associations” and families and participants who have experienced ABI.

2. Can the outcomes be explicitly be defined, e.g primary outcome is change from baseline to xxxxx. Similar comment for other outcomes.

3. Re sample size, please add details for rationale or source of ICC. And also the estimated number of sample of 4, is this per cluster?

4. Section on inclusion criteria can be moved to be in the methods section.

5. Define intention to treat population.

6. Appropriate statistical analysis would be a mixed effects model, especially a since you have a unit cluster, which may be needed to be treated as random effect.

7. Also how was missing data handled?

8. In table 2 – as this an RCT we do not perform formal comparisons of baseline characteristics, any differences occurring are by chance – so I would omit the p-values column

Reviewer #2: Abstract:

- The results need more details.

- The conclusion: add future recommendation "brief message" to readers and researchers.

Title:

- The short title needs to be shortened, for example "Web-based intervention and acquired brain injury"

Introduction:

- Add a clear hypothesis.

Methods:

- The study design, ethics, and setting are not clear.

- How and who administrates the data collection?

- How did you achieve the validity and reliability of the outcome measures?

- For statistical analysis, explain all methods used in detail.

- Please, re-frame the components (SPICES) for methods

i. Study design, setting, sample size

ii. Participants

iii. Intervention/issue of interest (exposure)

iv. Comparison

v. Ethics and end point

vi. Statistical analysis

- What were the eligibility criteria for participants?

- Mention the settings and locations where the data were collected.

- Provide sufficient details of interventions of each group to allow replication.

- Define pre-specified primary and secondary outcome measure.

- Explain with reasons any changes to study outcomes after the study commenced.

- How was sample size determined?

6. PLOS authors have the option to publish the peer review history of their article (what does this mean?). If published, this will include your full peer review and any attached files.

Reviewer #1: No

Reviewer #2: **Yes: **Walid Kamal Abdelbasset

---

## [Author Response · Author response to Decision Letter 0]

21 Jun 2022

Thank you very much for your e-mail regarding our manuscript, “Effectiveness of web-based intervention for life-change adaptation in family caregivers of community-dwelling individuals with acquired brain injury: A cluster-randomized controlled trial” (PONE-D-21-37509). We are grateful to know that it is potentially acceptable for publication in PLOS ONE. Please find attached a revised version of our manuscript.

Your comments and those of the reviewers were highly insightful and enabled us to greatly improve the quality of our manuscript. We include below our point-by-point responses to each of the comments of the reviewers as well as your own comments. 

We look forward to hearing from you regarding our re-submission. We would be happy to respond to any further questions and comments that you may have.

Respectfully yours,

Yuka Iwata (Shindo), MA, Etsuko Tadaka, PhD

Response to Reviewers

To the comments of Reviewer #1

1. The unit of clusters should be defined explicitly. It’s a bit confusing especially with the terminology of “associations” and families and participants who have experienced ABI. 

Response: We appreciate the reviewer’s suggestion. As suggested, we corrected lines 119–121 and fixed the related context.

Line119-121

“The unit of randomization is the institutions for families of individuals with ABI. All 82 institutions (cluster) for families of individuals with ABI in Japan were invited to participate in this study.”

　

Line 33-34

“Clusters were randomly assigned to the intervention (8 clusters, n = 120) or the control (8 clusters, n = 120) group.”

Line116-118

“Cluster randomization, instead of individual randomization, was adopted to reduce possible contamination among participants if an institution had both intervention and control group participants.”

2. Can the outcomes be explicitly be defined, e.g primary outcome is change from baseline to xxxxx. Similar comment for other outcomes. 

Response: We appreciate the reviewer’s suggestion. We observed outcomes at two time points: from baseline to 3 days as a short-term follow-up and from baseline to 1 month as a long-term follow-up. We have clarified the definition of the outcomes and have also reviewed all relevant texts.

Line 105-108

“The hypothesis was that, compared with the usual service, KF3 PGM will improve life-change adaptation in family caregivers of community-dwelling individuals with ABI at 3 days (short-term follow-up) and 1 month (long-term follow-up).”

Line 240-244

“The primary statistical hypothesis was that the LCAS scores would be improved in the intervention group compared with the control group at 3 days (short-term follow-up) and 1 month (long-term follow-up). The secondary statistical hypothesis was that the MSPSS and PAC scores would improve, as previously detailed.”

3. Re sample size, please add details for rationale or source of ICC. And also the estimated number of sample of 4, is this per cluster? 

Response: We have added a detail for the ICC rationale. The estimated number of samples (i.e., 4) is per cluster. We have added the following text (lines 137–146):

Line 137-146

“The sample size based on the following expectation was used. We powered the trial to have 80% power (alpha = 0.05; intraclass correlation coefficient (ICC) = 0.04) to detect a difference between groups on the primary outcome at follow-up to a standardized effect size (delta = 3.0; SD = 4.3). Allowing for 20% attrition, the estimated sample size was 4 participants per cluster; the required cluster size was 8 clusters per group. Our forecast was principally based on our pilot study findings (refer to S1 and S2 Study protocol). The ICC was principally based on the findings of previous study estimates [21]. The number of participants per cluster was principally based on the findings of a previous study of family caregivers of individuals with ABI in Japan, where the average recruitment in the previous study was 4 participants per cluster [13].”

4. Section on inclusion criteria can be moved to be in the methods section.

Response: As suggested, the section on inclusion criteria has been moved to the participant subsection of the methods section.

5. Define intention to treat population. 

Response: We appreciate the reviewer’s suggestion. We have corrected Fig. 2 and have added the following text:

Line 265-268

"A total of 91 participants (intervention group n = 41; control group n = 50) were included in the intention-to-treat analysis (Fig 2). A total of 67 participants completed the study (retention rate: 26.4%): 24 participants withdrew by the 1 month interval (intervention group n = 10; control group n = 14)."

6. Appropriate statistical analysis would be a mixed effects model, especially a since you have a unit cluster, which may be needed to be treated as random effect.

Response: We agree the reviewer’s suggestion. We have analyzed using the mixed model for repeated measures. Therefore we have revised the analysis explanation in this context, the result, discussion and table 2.

Method― Line 39-40

“A mixed model for repeated measures (MMRM) was applied.”

Method― Line 245-254

“Descriptive statistics such as the mean, standard deviation, and frequency distribution were used to describe the demographic characteristics of the participants. A mixed model for repeated measures (MMRM), which takes into account the multiple measurements (3 time points: baseline, short-, and long-term follow-ups), was performed. MMRM was used to test a group × time interaction. Fixed effects included 1) group allocation (intervention or control), 2) time, and 3) group × time interaction; random effects included cluster. Analyses were performed on an intention-to-treat basis and included all baseline respondents. Missing data were not explicitly imputed because MMRM applies with a restricted maximum likelihood solution to repeated measures analyses under the missing-at-random assumption.”

Result― Line 42-47

“A total of 91 participants were enrolled. The mean age (SE) of the participants was 64.0 (9.2) years; 87.8% of them were female. The intervention group showed better improvement in the LCAS than the control group in the whole study period (F = 6.5, p = 0.002). The mean observed change in LCAS from baseline was +8.0 (SE = 2.0) at 3 days and +11.6 (SE = 2.0) at 1 month in the intervention group (F = 18.7, p < 0.001). No significant differences in MSPSS and PAC were observed among the intervention and control groups in the whole study.”

Result― Line 297, Table 2

Discussion― Line 324-325

“The intervention group participants exhibited better life-change adaptation than the control group in the whole study period.”

Discussion― Line 332-337

“Grounded in the model [20], the key to enhancing the adaptation was three domains: social support, appraisals, and coping [30–32]. In the case of social support, the MSPSS scores were not significantly different between groups. A previous study suggested that just involvement with the program can increase self-efficacy of attribution in the program population [33]. What both the intervention and control groups provide a family institution-based routine program might not affect each group in MSPSS.”

Discussion― Line 340-343

“However, eco-mapping in the KF3 PGM is a tool that encourages reflection on the perceived social support [35,36]. Thus, we found that the KF3 PGM served as a domain of social support that could facilitate life-change adaptation.”

Discussion― Line 343-347

“In the case of appraisals, the PAC score improved from baseline to short- and long-term follow-up.” “The KF3 PGM provides a message board facilitating interpersonal interaction. Thus, we found that the KF3 PGM served as a domain of appraisal that could facilitate life-change adaptation. “

7. Also how was missing data handled?

Response: Because we used the mixed model for analyses, missing data were not explicitly imputed. We have added an explanation in this context.

Line 252-254

"Missing data were not explicitly imputed because MMRM applies with a restricted maximum likelihood solution to repeated measures analyses under the missing-at-random assumption."

8. In table 2 – as this an RCT we do not perform formal comparisons of baseline characteristics, any differences occurring are by chance – so I would omit the p-values column.

Response: As suggested, we have omitted the p-values column in Table 2 (now Table 1). We have also removed the text comparing baseline characteristics.

Line 272-273

“Table 1 shows participants’ demographic characteristics at baseline in each group.”

 

To the comments of Reviewer #2

1. Abstract: －The results need more details.

Response: We agree with the reviewer’s suggestion. Given the points suggested by other reviewers, we have changed the analysis method to a mixed model for repeated measures (MMRM) and have revised the details accordingly. 

Line 42-47

“A total of 91 participants were enrolled. The mean age (SE) of the participants was 64.0 (9.2) years; 87.8% of them were female. The intervention group showed better improvement in the LCAS than the control group in the whole study period (F = 6.5, p = 0.002). The mean observed change in LCAS from baseline was +8.0 (SE = 2.0) at 3 days and +11.6 (SE = 2.0) at 1 month in the intervention group (F = 18.7, p < 0.001). No significant differences in MSPSS and PAC were observed among the intervention and control groups in the whole study.”

2. Abstract: －The conclusion: add future recommendation "brief message" to readers and researchers.

Response: We appreciate the reviewer’s suggestion. We have added a brief message (lines 50–54). We have also appended the relevant sections for consistency in the paper (lines 315–323 and 362–370).

Line50-54

“The results show that a potential web-based intervention in institutions for families of individuals with ABI plays a substantial, longer-term role in their support in Japan. Future studies could address the same research questions in different settings and cultures for family caregivers for even longer time periods.”

Line 315-323

“The originality of this study is the use of cluster-randomization and measurement of the life-change adaptation by LCAS. The findings show that a potential web-based intervention in institutions for families of individuals with ABI plays a substantial, longer-term role in their support. That is, the unified web-based intervention in an institutional program has the potential to strengthen said institutions, which are essentially self-help groups, and enhance life-change adaptation as an ABI family. In addition, this study is the first step toward looking at a self-help group that can build on caregivers’ strengths such as life-change adaptation and examining a practicable unified program for a self-help group.”

Line 362-370

“The first limitation concerns the study setting and/or culture; convenience sampling of institutions in Japan, rather than probability sampling in other settings and/or cultures, was used. In generalizing the findings of this study, one must consider that the study was conducted through institutions of family caregivers in Japan, where family caregiving and family ties are traditionally and relatively more important than in other cultures.“ ”Accordingly, future studies could address the same research questions for family caregivers in different settings and cultures and for even longer time periods.”

3. Title: －The short title needs to be shortened, for example "Web-based intervention and acquired brain injury"

Response: As suggested, we have corrected line 8.

Line 8

“Web-based intervention for life-change adaptation”

4. Introduction: －Add a clear hypothesis.

Response: We have added a hypothesis in the introduction and a statistical hypothesis in the methods section.

Line 105-108

“The hypothesis was that, compared with the usual service, KF3 PGM will improve life-change adaptation in family caregivers of community-dwelling individuals with ABI at 3 days (short-term follow-up) and 1 month (long-term follow-up).”

Line 240-244

“The primary statistical hypothesis was that the LCAS scores would be improved in the intervention group compared with the control group at 3 days (short-term follow-up) and 1 month (long-term follow-up). The secondary statistical hypothesis was that the MSPSS and PAC scores would improve, as previously detailed.”

5. Methods: －The study design, ethics, and setting are not clear

Response: As suggested, we have added a careful discussion of the study design, ethics, and setting. Following the reviewer’s suggestion #9, we have also added a discussion of the SPICE framework.

Line 151-153

“Of the 18 eligible clusters, studies conducted in 16 randomly selected clusters were performed on the basis of the anticipated sample size. A total of 16 clusters and 240 members were enrolled in the study.”

Line 226-230

“The informed consent document explained the voluntary nature of participation, management of data, and intention to publish the results. All participants received the informed consent document and were considered to have agreed to participate in this study if they made a check mark on the informed consent document.”

Line 258-261

“Each cluster provided service to a target within their catchment area, and participants selected in the control and intervention groups were far from each other, enabling us to avoid information contamination.”

6. Methods: －How and who administrates the data collection?

Response: As suggested, we have added descriptions of the data collection and administration methods.

Line 127-136

“The clusters mailed potential participants sequential invitations to express interest in joining the study. Both the intervention and control groups’ participants were given a questionnaire for the survey via mail. All participants completed questionnaires at home. The survey was examined at three points in time: baseline, 3 days (short-term follow-up), and 1 month (long-term follow-up). Baseline data were sent to the research institute (Department of Community Health Nursing, Yokohama City University; YCU) when the participant enrolled and completed the survey. In addition, short- and long-term follow-up data were sent to YCU by the end of the study. The collected data were entered by independent research assistants and managed by YCU.”

7. Methods: －How did you achieve the validity and reliability of the outcome measures?

Response: As suggested, we have revised the section on outcomes.

Line 202-205

“The reliability of the scale has been established (Cronbach’s α: 0.84), and validity was confirmed for Japanese participants in a previous study [13]. The Cronbach’s α in the present study was 0.84 at baseline, 0.86 at 3 days, and 0.88 at 1 month.” 

Line 211-213

“The reliability of the scale has been established (Cronbach's α: 0.91), and the validity was confirmed for Japanese participants in a previous study [25]. The Cronbach's α in the present study was 0.93 at baseline, 0.92 at 3 days, and 0.95 at 1 month.”

Line 217-219

“The reliability of the scale has been established (Cronbach's α: 0.92), and the validity was confirmed for Japanese participants in a previous study [27]. The Cronbach's α in the present study was 0.91 at baseline, 0.91 at 3 days, and 0.90 at 1 month.”

8. Methods: －For statistical analysis, explain all methods used in detail.

Response: As suggested, we added descriptions of all the methods in the “Statistical analysis” section. Given the points suggested by other reviewers, we have removed the analysis for the chi-square test and independent t-test. We have also changed the analysis method to a mixed model for repeated measures (MMRM). 

Line 245-254

“Descriptive statistics such as the mean, standard deviation, and frequency distribution were used to describe the demographic characteristics of the participants. A mixed model for repeated measures (MMRM), which takes into account the multiple measurements (3 time points: baseline, short-, and long-term follow-ups), was performed. MMRM was used to test a group × time interaction. Fixed effects included 1) group allocation (intervention or control), 2) time, and 3) group × time interaction; random effects included cluster. Analyses were performed on an intention-to-treat basis and included all baseline respondents. Missing data were not explicitly imputed because MMRM applies with a restricted maximum likelihood solution to repeated measures analyses under the missing-at-random assumption.”

9. Methods:－Please, re-frame the components (SPICES) for methods; i. Study design, setting, sample size, ii. Participants, iii. Intervention/issue of interest (exposure), iv. Comparison, v. Ethics and end point, vi. Statistical analysis

Response: Thank you for these suggestions to improve the presentation of our study. We agree with most of these remarks. We have provided the required details and have revised the methods section according to the reviewers’ comments.

10. Methods:－What were the eligibility criteria for participants?

Response: The following has been added to the manuscript. In addition to a description of the criteria for participants, a description of the criteria for clusters was added.

Line 155-160

“Participants’ inclusion criteria were (1) caring for an individual with ABI, (2) the caregiver was aged 20 years or more, (3) the individual with ABI developed ABI when aged more than 16 years and less than 65 years. The reasons for excluding cases in which ABI occurred in an individual aged more than 16 years or less than 65 years are that the adaptation process and outcomes for ABI are different[13]. Participants’ exclusion criteria were (1) withdrew consent for data use and (2) missing LCAS baseline data.”

Line 150-151

“Clusters’ exclusion criteria were (1) less than 15 members, (2) unspecified members, and (3) refusals of explanations.”

11. Methods: －Mention the settings and locations where the data were collected.

Response: As suggested, we have mentioned the setting and locations.

Line 129-136

“All participants completed questionnaires at home. The survey was examined at three points in time: baseline, 3 days (short-term follow-up), and 1 month (long-term follow-up). Baseline data were sent to the research institute (Department of Community Health Nursing, Yokohama City University; YCU) when the participant enrolled and completed the survey. In addition, short- and long-term follow-up data were sent to YCU by the end of the study. The collected data were entered by independent research assistants and managed by YCU.”

12. Methods:－Provide sufficient details of interventions of each group to allow replication.

Response: Thank you for these suggestions. We have revised the “Intervention” and “Comparison” section and have added Figure 1.

Line 180-183

“The 3 consecutive days of the KF3 PGM had three components: (1) a practical guide part, (2) an eco-mapping part, and (3) a message-board part. KF3 PGM is described in Fig 1. After accessing the website, the participants were presented with three components.”

Line 192-195

“The control group participants were guided in their daily routine. They were free to receive usual services such as consultation and information regarding ABI from institutions to which they belong. They were then placed on a waiting list to receive the intervention after 1 month.”

13. Methods:－ Define pre-specified primary and secondary outcome measure.

Response: As suggested, we have revised the outcomes section.

Line 199-201

“Life-change adaptation” refers to “the outcome of adaptation to changes in living resources / health belief of life of family caregivers due to ABI” [13].

Line 208-210

“Social support” refers to “an exchange of resources between at least two individuals perceived by the provider or recipient to be intended to enhance the well-being of the recipient” [23].

Line 214-216

“Positive appraisal of care” refers to “any cognitive and affective evaluation of one’s caregiving experience that describes caregiving as good and comfortable” [27].

14. Methods: － Explain with reasons any changes to study outcomes after the study commenced.

Response: There were no changes to the outcomes measured in the study design after the study commenced.

15. Methods: － How was sample size determined?

Response: As suggested, we have carefully revised the details of the sample size.

Line137-146

“The sample size based on the following expectation was used. We powered the trial to have 80% power (alpha = 0.05; intraclass correlation coefficient (ICC) = 0.04) to detect a difference between groups on the primary outcome at follow-up to a standardized effect size (delta = 3.0; SD = 4.3). Allowing for 20% attrition, the estimated sample size was 4 participants per cluster; the required cluster size was 8 clusters per group. Our forecast was principally based on our pilot study findings (refer to S1 and S2 Study protocol). The ICC was principally based on the findings of previous study estimates [21]. The number of participants per cluster was principally based on the findings of a previous study of family caregivers of individuals with ABI in Japan, where the average recruitment in the previous study was 4 participants per cluster [13].”

Editorial Requests:

Response: We appreciate the advice and have used the PLOS ONE style templates to prepare our revised manuscript.

Response: We have added the requested information. Our study did not include minors.

Line 226-230

“The informed consent document explained the voluntary nature of participation, management of data, and intention to publish the results. All participants received the informed consent document and were considered to have agreed to participate in this study if they made a check mark on the informed consent document.” 

Response: In accordance with feedback from Request No. 4, we have removed the "Funding Information" section.

This work was supported by the Sasakawa Scientific Research Grant from the Japan Science Society. 

Response: Thank you for the suggestion. Having removed funding-related text from the manuscript, we would like to add the funding information about "the Sasakawa Scientific Research Grant from the Japan Science Society" to the final published paper. We would like to update your Funding Statement as follows: 

“This work was supported by the Sasakawa Scientific Research Grant from the Japan Science Society (grant number: 2021-6012). The funders had no role in study design, data collection and analysis, decision to publish, or preparation of the manuscript.”

Response: Publication of the dataset for this study qualifies as "b)." We have uploaded the minimal anonymized data set as Supporting Information file "S4_Data set."

---

## [Decision Letter · Decision Letter 1]

8 Aug 2022

Effectiveness of web-based intervention for life-change adaptation in family caregivers of community-dwelling individuals with acquired brain injury: A cluster-randomized controlled trial

PONE-D-21-37509R1

Dear Dr. Iwata(Shindo),

We’re pleased to inform you that your manuscript has been judged scientifically suitable for publication and will be formally accepted for publication once it meets all outstanding technical requirements.

Kind regards,

Andrea Martinuzzi

Academic Editor

PLOS ONE

Additional Editor Comments (optional):

Reviewers' comments:

Reviewer's Responses to Questions

**Comments to the Author**

1. If the authors have adequately addressed your comments raised in a previous round of review and you feel that this manuscript is now acceptable for publication, you may indicate that here to bypass the “Comments to the Author” section, enter your conflict of interest statement in the “Confidential to Editor” section, and submit your "Accept" recommendation.

Reviewer #1: All comments have been addressed

Reviewer #2: All comments have been addressed

2. Is the manuscript technically sound, and do the data support the conclusions?

Reviewer #1: Yes

Reviewer #2: Yes

3. Has the statistical analysis been performed appropriately and rigorously? 

Reviewer #1: Yes

Reviewer #2: Yes

4. Have the authors made all data underlying the findings in their manuscript fully available?

Reviewer #1: No

Reviewer #2: Yes

5. Is the manuscript presented in an intelligible fashion and written in standard English?

Reviewer #1: Yes

Reviewer #2: Yes

6. Review Comments to the Author

Reviewer #1: (No Response)

Reviewer #2: Effectiveness of web-based intervention for life-change adaptation in family caregivers of community-dwelling individuals with acquired brain injury: A cluster-randomized controlled trial

No further comments are required.

7. PLOS authors have the option to publish the peer review history of their article (what does this mean?). If published, this will include your full peer review and any attached files.

Reviewer #1: No

Reviewer #2: **Yes: **Walid Kamal Abdelbasset

---

## [Editor Report · Acceptance letter]

10 Aug 2022

PONE-D-21-37509R1 

Effectiveness of web-based intervention for life-change adaptation in family caregivers of community-dwelling individuals with acquired brain injury: A cluster-randomized controlled trial 

Dear Dr. Iwata(Shindo):

I'm pleased to inform you that your manuscript has been deemed suitable for publication in PLOS ONE. Congratulations! Your manuscript is now with our production department. 

Kind regards, 

on behalf of

Dr. Andrea Martinuzzi 

Academic Editor

PLOS ONE